# Immunotherapy as a Promising Option for the Treatment of Advanced Chordoma: A Systemic Review

**DOI:** 10.3390/cancers15010264

**Published:** 2022-12-30

**Authors:** Xiang Wang, Zhaoyu Chen, Bo Li, Jiefu Fan, Wei Xu, Jianru Xiao

**Affiliations:** Department of Orthopedic Oncology, Changzheng Hospital, Second Military Medical University, #415 Fengyang Road, Shanghai 200003, China

**Keywords:** chordoma, immunotherapy, vaccines, immune checkpoint inhibitor, combination

## Abstract

**Simple Summary:**

Chordoma is a rare orthopedic tumor that is mainly treated with surgery and radiotherapy. Due to the disease characteristics of chordoma, it is difficult to remove completely and easily recur. For patients with advanced chordoma who have failed in surgery and radiotherapy, immunotherapy may be a proposing option. We aimed to assess the efficacy and safety of different immunotherapy agents in people with advanced chordoma. We concluded that immune checkpoint inhibitors are the most effective therapies. Because the combination therapy of immune checkpoint inhibitors more likely to cause adverse event, as well as there is no evidence that it is superior to monotherapy, to a greater extent in clinical practice, monotherapy of anti-PD-1/PD-L1 inhibitors is recommended. Pembrolizumab is the most commonly used monotherapy. Tumor vaccines are the safest immunotherapy drugs for patients, but meanwhile their efficacy is inferior to immune checkpoint inhibitors. Immunomodulatory drugs are not recommended to be preferred due to their unremarkable efficacy and inadequate safety. These conclusions may be serviceable for people with advanced chordoma and their physicians in choosing an appropriate immunodrug for their treatment.

**Abstract:**

Objective: To summarize the function and efficacy of immunotherapy as an adjunctive therapy in the treatment of advanced chordoma. Methods: Literature search was conducted by two reviewers independently. Case reports, case series and clinical trials of immunotherapy for chordoma were retrieved systematically from Pubmed, Web of Science, Scoupus and Cochrane Library. Clinical outcome data extracted from the literature included median progression-free survival (PFS), median overall survival (OS), clinical responses and adverse events (AEs). Results: All studies were published between 2015 and 2022. Twenty-two eligible studies were selected for systemic review. PD-1/PD-L1 immune checkpoint inhibitors (ICIs) were the most common used immunotherapy agents in chordoma, among which Pembrolizumab was the most frequently prescribed. CTLA-4 antibody was only used as combination therapy in chordoma. Dose Limiting Toxicity (DLT) was not observed in any vaccine targeting brachyury, and injection site response was the most frequent AV. The response evaluation criteria in solid tumors (RECIST) were the most generally used evaluation standard in chordoma immunotherapy, and none of the included studies employed the Choi criteria. Conclusions: No clinical data have demonstrated that CTLA-4 ICIs combined with PD-1/PD-L1 ICIs is more effective than ICIs monotherapy in treating chordoma, and ICIs in combination with other therapies exhibit more toxicity than monotherapy. PD-1/PD-L1 ICIs monotherapy is recommended as an immunotherapy in patients with advanced chordoma, which may even benefit PD-L1-negative patients. The brachyury vaccine has shown good safety in chordoma patients, and future clinical trials should focus on how to improve its therapeutic efficacy. The use of immunomodulatory agents is a promising therapeutic option, though additional clinical trials are required to evaluate their safety and effectiveness. RECIST does not seem to be an appropriate standard for assessing medications of intratumoral immunotherapy.

## 1. Introduction

Chordoma is a malignant tumor originating from the embryonic notochord and often distributed along the axial skeleton [1], with an annual incidence of 0.08 per 100,000/year [2,3,4]. It can develop anywhere along the spinal column with a predilection for the clivus and sacrum [3]. Owing to its unique growth location, chordoma often involves important nerves and blood vessels, making clinical management more difficult [5]. En bloc resection is the treatment of choice to prolong the survival of chordoma patients due to negative microscopic margins. If en bloc resection is possible, post-lesion resection paired with radiotherapy should not be considered as an option [6,7,8]. Complete radical resection yields higher local control rates and prolongs local recurrence-free survival (RFS) as compared with subtotal resection [9,10]. However, when a patient becomes symptomatic, the tumor has often involved the surrounding vital tissues, making complete excision almost impossible [11]. Most patients with chordoma develop local recurrence. In advanced stages of chordoma, surgical intervention become difficult and additional systemic therapy is often required [12]. The US Food and Drug Administration (FDA) has approved nonmedical treatment for advanced chordoma in chemotherapy resistant patients. Therefore, there is an urgent need to find effective agents for treating chordoma [13].

Although immunotherapy is not the mainstay of treatment for chordoma, its application potential in chordoma has been explored in recent years, including brachyury vaccines and immune checkpoint inhibitors, which have provided more treatment options for this rare disease [14]. However, there is no systemic review about clinicians’ preference for immunotherapy, as well as safety, therapeutic efficacy and the common evaluation criteria of chordoma immunotherapy. The objective of this systemic review is to investigate the function and efficacy of immunotherapy as an adjunctive therapy in the treatment of advanced chordoma.

## 2. Materials and Methods

The present systematic review was performed based on PRISMA guidelines, and registered in PROSPERO (registration number: 381775). The research was conducted using PubMed, Scoupus, Cochrane and Web of Science, using following strings: ((((((immune) OR (immunotherapy)) OR (vaccine)) OR (ICI) OR (Immune checkpoint inhibitor) OR (“cell therapy”)) AND (chordoma). There was no limit on the search strategy.

Studies associated with immunotherapy on chordoma, including clinical trials, case reports and case series related to chordoma immunotherapy were selected for inclusion. Case series, case reports, and clinical trials published solely in abstracts were analyzed only when they contained new data. New clinical trials for chordoma were found from the Chordoma Foundation, ClinicalTrials.gov, EU Clinical Trials Register. The exclusion of studies was based on the following criteria: (a) the publications were not in English; (b) the articles were unavailable to our institution; (c) studies examining the effect of immunotherapy on cancers other than chordoma; and (d) studies without evaluating the number of patients with chordoma and the use of drugs.

Two authors (WX, CZY) independently performed the search, selected the publications, and retrieved data from each study. Disputes were addressed with the assistance of a third author (LB). The following data were extracted: the type of publication (study design and publication year), patient characteristics (number of patients, treatment history, site of tumor), immunotherapy characteristics (type of drugs and adverse events (AEs)), clinical response evaluation criteria (Choi’s criteria and RECIST), median PFS and median OS.

## 3. Results 

### 3.1. Search Results

A total of 329 articles were identified in the initial screening, of which 280 were excluded after reading the abstracts. Of the remaining 49 articles, 20 met our inclusion criteria, and additional two were found eligible by manual research of the reference list of the 20 included articles. Finally, 22 article were included for this systemic review. The flowchart for the exclusion and inclusion of related studies is shown in Figure 1. These publications categorize ICIs, vaccines and other immunomodulatory drugs, according to the type of immunotherapy. 

### 3.2. Study Characteristics

Among the 22 studies, 13 were clinical trials [15,16,17,18,19,20,21,22,23,24,25,26,27], including three retrospective case series [28,29,30], and six case reports [31,32,33,34,35,36] (Figure 2). Pembrolizumab was evaluated in five studies involving 46 patients [16,28,33,34,36], Nivolumab was evaluated in five studies involving six patients [28,29,32,34,35], MVA (Modified Vaccinia Ankara)-brachyury vaccine was evaluated in three studies involving 26 patients [17,20,23]. In addition, Yeast-brachyury vaccine (GI-6301) was evaluated in two studies with 21 patients, Durvalumab in two studies with six patients, and Ipilimumab in two studies with two patients [18,24,25,28,30,32,37], tremelimumab in five patients, HuMax-IL8 in five patients, FAZ053 in two patients, sintilimab in one patient, PG545 in one patient, avelumab in one patient, Clostridium novyi-NT (non-toxic; lacking the alpha toxin) in one patient, HSV1716 in one patient, and MVX-ONCO-1 in one patient in one study each [15,21,22,26,27,28,31,34]. PD-1/PD-L1 ICIs was evaluated in 12 studies involving 67 patients [16,22,28,29,30,31,32,33,34,35,36,37], CTLA-4 ICIs in three studies involving seven patients [25,28,32,37], Brachyury vaccine in five studies involving 47 patients [17,18,20,23,24]. Other immunomodulatory drugs, including HuMax-IL8, PG545, C. novyi NT spores, HSV1716, MVX-ONCO-1, were analyzed in five studies involving nine patients [15,21,26,27,28,34]. Monotherapy of immunotherapy was reported in 15 studies [15,16,17,20,21,22,23,24,26,27,28,30,32,33,34,35], and with combination therapy in six studies [18,25,28,29,31,32]. The RECIST evaluation criteria were applied in 14 studies [15,16,17,18,20,21,22,23,24,26,27,28,32,35], and none of eligible studies used Choi’s criteria. AEs were reported in 19 studies, including 3–4 AEs such as myocarditis, colitis and pneumonitis, as well as 1–2 AEs such as injection-site reaction, fatigue, fever, nausea and vomiting [15,16,17,18,20,21,22,23,24,25,26,27,28,29,30,31,33,34,37].

### 3.3. Efficacy and Safety of Immunotherapy in Chordoma Patients 

The overall response rate (ORR) is defined as complete response (CR) or partial response (PR) at the tumor site. ICIs were evaluated by RECIST in five studies involving 53 patients, with an ORR of 15% (*n* = 8) [16,28,35,36,37]. Three studies reported grade 3–4 AEs [16,28,37]. Brachyury vaccines were evaluated by RECIST in four clinical trials involving 35 patients [17,18,20,24], with an ORR of 8% (*n* = 3). No DLT was observed and grade 3 AEs were reported in one study [17]. Immunomodulatory drugs were analyzed by RECIST in four studies involving eight patients, and none of them achieved CR or PR, with an ORR of 0% [15,21,26,27]. DLT was observed in two studies [21,26]. ICIs were the most effective immunotherapy drugs for chordoma and the brachyury vaccine enjoyed the optimal safety profile, though immunomodulatory drugs required more exploration and improvement.

### 3.4. PD-1/PD-L1 Immune Checkpoint Inhibitors

Pembrolizumab is an immune checkpoint inhibitor targeting PD-1, which was the most widely used ICIs in patients with chordoma. The therapeutic efficacy of Pembrolizumab was evaluated in five studies involving 46 patients, including one clinical trials [16], one case series [28], and three case reports [33,34,36] (Table 1). All the studies used Pembrolizumab as monotherapy, including three studies and 44 patients used RECIST [16,28,36]. Three studies reported PFS and OS [16,28,34]. Massard et al. conducted a phase II trial involving 34 chordoma patients, of whom three patients achieved PR with a median PFS of 9 months and 1-year OS of 87%. However, the study did not report the number of chordoma patients with stable disease (SD) or progressive disease (PD) [16]. In a retrospective case series with nine chordoma patients, three patients achieved CR or PR, and six patients achieved SD. The median PFS was 6 months, and no median OS was reached [28]. A case report with one chordoma patient reported that the tumor was controlled for six months [34]. AEs were reported in four studies of 13 patients [28,33,34,36], including four patients (33.3%) who discontinued the use of ICIs because of toxicities including colitis, myocarditis, pneumonitis and panhypopituitarism [28,33]. 

Nivolumab is a PD-1 antibody, which was evaluated in five studies, including three case reports and two retrospective case series [28,29,32,34,35]. Nivolumab monotherapy was employed in three studies (three patients) [32,34,35], including one study that evaluated the treatment by RECIST and reported PR in one patient [35]. No complications were observed under nivolumab and pazopanib [29].

Durvalumab, an immune checkpoint inhibitor targeting PD-L1, was analyzed in two studies [30,37]. In a phase II clinical trial using RECIST to evaluate Durvalumab plus tremelimumab in five chordoma patients, the authors reported PR in one patient, SD in three patients, and PD in one patient [37]. AEs included Sicca syndrome, Maculopapular rash and Pruritus [28,30].

FAZ053, an ICIs against PD-L1, was evaluated in a retrospective case series, reporting SD in two patients [28]. Avelumab, a PD-L1 antibody, was assessed in a phase I trial with one patient. This patient experienced grade 1/2 AEs [22]. Sinitilimab, a third line treatment for chordoma, was assessed in a case report combined with anlotinib. The patient suffered ICIs-associated AEs including myocarditis and immune-related hepatitis. He was treated with methylprednisolone and died within a week despite AEs being controlled [31]. 

### 3.5. CTLA-4 Immune Checkpoint Inhibitor

All immunotherapies using CTLA-4 Immune checkpoint inhibitor were combination therapies [25,32,37]. AEs were reported in two studies [25,37] (Table 2).

Ipilumimab, a CTLA-4 antibody, was assessed in two studies [25,32], neither of which used RECIST. A phase I/II trial investigated the efficacy of Intratumoral INT230-6 combined with Ipilumimab in a chordoma patient. The patient achieved disease control. Most AEs were of low grade and transient without related grade 4 or 5 toxicities, while three patients developed Ipilumimab-associted serious adverse events (SAEs) such as colitis [25]. Another study evaluating Nivolumab combined with Ipilumimab reported CR for more than two years in a patient with metastatic lung lesions, while the primary sacral chordoma remained resistant to immunotherapy [32]. 

Tremelimumab, another CTLA-4 antibody, was analyzed in one study in combination with durvalumab, described in the Durvalumab section above [28]. The most frequent severe AE was colitis, and the other serious AEs included pneumonitis, abdominal pain and myocarditis. 

### 3.6. Cancer Vaccine 

MVA-brachyury, a MVA vector-based vaccine expressing the transgenes for brachyury, was evaluated in three studies with 27 patients [17,20,23], of which two studies used RECIST [17,20], reporting PR in one patient (7%), SD in six patients (46%) and PD six patients (46%) (Table 3). A phase I study reported that the median PFS was 253 days [20]. The most frequent AE was injection-site reaction, followed by fever and fatigue. In all phase I trials, no dose-limiting toxicity (DLT) was observed, and all AEs related to the vaccine were grade 1 or 2 [17,20,23]. An MVA-brachyury-TRICOM vaccine that encodes Brachyury expression and three human T-cell costimulatory molecules, was used to stimulate an increased immune response against brachyury. In a phase I clinical trial including 11 patients, 10 developed brachyury-specific T-cell responses, demonstrating that the treatment was safe with specific immunogenicity, though immune responses were not durable [38]. A subsequent phase I clinical trial added an anipox vector enhancer to the MVA vaccine to further enhance brachyury-specific T-cell responses in three patients with advanced chordoma, the result showed that the best response was SD according to RECIST, and the resulting immune response continued to increase [17].

In a phase 1 open-label trial of MVA-brachyury-TRICOM vaccine involving 10 patients with advanced chordoma, impressive clinical responses were observed in two patients, including one who achieved PR for more than 10 months, and the other who achieved sustainable SD according to RECIST. Although the patient with SD had a 2.7% reduction in tumor length, he achieved a 41% reduction in tumor volume and experienced meaningful and lasting clinical improvement.

Yeast-Brachyury Vaccine (GI-6301) was evaluated in two studies [19,24]. Of the 19 patients evaluated by RECIST, two (10%) achieved PR, eight (42%) achieved sustainable SD, eight (42%) achieved PD, and one patient (5%) achieved mixed response (MR). A patients received Yeast-Brachyury Vaccine regimen after radiotherapy, in whom the tumor size at the previous radiotherapy site was significantly reduced, but tumor growth was observed in an area without receiving irradiation. According to the RECIST, his tumor measurement was generally stable, which is considered a “MR” [24]. In a phase II study evaluating the effectiveness of radiation combined with yeast-Brachyury Vaccine, the median PFS was 20.6 months [18]. In a clinical trial evaluating the yeast-brachyury GI-6301 vaccine in patients with advanced chordoma, one patient achieved PR over 600 days according to RECIS, and the tumor shrank for more than 30% in another patient. Both patients received radiotherapy before enrollment [39]. Yeast-Brachyury Vaccine was well tolerated. No DLT was observed, and the main AE was injection-site reaction, followed by reduced lymphocyte count and fever [18,24].

### 3.7. Other Immunomodulatory Drugs

HuMax-IL8 is a monoclonal antibody targeting interleukin-8 (IL-8), which was evaluated in a phase I trial (NCT02536469) with five patients [15], of whom four patients achieved SD and the other patient achieved PD according to RECIST, with a median PFS of 5.5 months. No DLT was observed, and the common AE was nausea (Table 4).

PG545 is an immunomodulatory agent that inhibits tumor-associated macrophages (TAM), which was evaluated in a phase I trial (NCT02042781) [21], reporting SD in one patient according to RECIST. The most frequent AE was hypertension, and DLTs were hypertension and epistaxis.

MVX-ONCO-1 is a personalized cell-based cancer immunotherapy, which was evaluated in a phase I trial [34]. The patient’s tumor was controlled with no relapse for over 90 months. No systemic toxicity was observed.

Clostridium novyi-NT (C. novyi-NT) is an attenuated strain of C. novyi, obligate anaerobe that lacks the lethal alpha toxin. A phase I trial analyzed the effect of this oncolytic bacteria (NCT01924689) [26], revealing that one patient achieved SD according to RECIST. DLTs were grade 4 sepsis and grade 4 gas gangrene, and other treatment-related grade toxicities included limb abscess and soft-tissue infection.

HSV1716 is an oncolytic herpes simplex virus-1 (HSV-1), which was reported in a phase I study [27]. A pediatric patient achieved SD for less than one month and died in 2.5 months. The study used modified RECIST to measure the longest diameter instead of the sum of the longest diameters. No DLT was observed, and the common AEs included fever, chills, and mild laboratory abnormalities such as anemia and leukopenia. Clinical trials programs of chordomas in progress are exhibited in Table 5.

## 4. Discussion

### 4.1. Indications and Evaluation Criteria for Immunotherapy

Immunotherapy is only suggested for patients with advanced chordoma that is refractory to surgical excision and radiation.

The Choi’s criteria have been applied to evaluate targeted therapy for chordoma, but not for immunotherapy. The reason may be that targeted drugs such as Imatinib lead to tumor necrosis and cystic degeneration with little change in tumor size as compared with immunotherapy such as ICIs [40]. Although RECIST is the most generally utilized criteria for evaluating immunotherapy of chordoma, it may not be a reliable indicator of clinical benefits with intratumoral immunotherapy agents [25]. Responses defined by RECIST depend on changes in the target lesion size assessed by noninvasive imaging. The amount of intratumoral drug administration and that retained in the tumor, even spore germination in oncolytic bacterial agents can complicate the evaluation of tumor growth [26]. In a clinical trial investigating intratumoral oncolytic bacteria, 19 patients achieved SD according to RECIST, and significant reduction in tumor density (−4% to −476%) was observed in 11 of these patients according to the Choi’s criteria [26]. Although no Choi’s criteria were employed in any immunotherapy research as evaluation criteria, they may be a better surrogate for evaluating intratumoral drugs.

### 4.2. Cancer Vaccines

Brachyury is a transcription factor belonging to the T-box gene family. Brachyury overexpression is exclusive to chordoma in comparison to other malignancies [41]. Several studies have demonstrated that Brachyury overexpression is closely related to poor prognosis and shorter PFS of patients [13,42,43]. In vitro inhibition of Brachyury expression led to growth arrest and apoptosis in chordoma cells [44]. Brachyury vaccine shows a promising perspective in chordoma. The safety and tolerability of brachyury vaccines are believed to be higher than ICIs and other immunomodulatory drugs in chordoma patients. No severe AEs were observed in any clinical trials, with injection-site reaction being the most frequent adverse event. The next step should be to place greater emphasis on measures to improve the efficacy of the vaccines, such as the use of combination therapies or improving the ability to activate Brachyury-specific T cells.

An preclinical study demonstrated the potential of brachyury vaccine combined with ICIs therapy [13]. Notably, both patients who achieved impressive clinical improvement were treated with ICIs prior to the use of MVA-brachyury-TRICOM vaccine [45], suggesting that the brachyury vaccine combined with ICIs may prove to be a potential regimen for chordoma patients.

Vaccines only exert clinical effects on patients with tumors expressing HLA class I antigens, and radiotherapy enhances the expression of HLA class I antigens on tumors, which provides a basis for combining vaccine therapy with radiation therapy [14]. A phase I trial reported that two patients who received radiation therapy prior to receiving the GI-3601 vaccine achieved significant clinical improvement [24]. These results suggest a synergistic effect between radiotherapy and vaccines, though more clinical trials are required to verify the conclusion.

In 2022, a phase II clinical trial of radiation combination with a vaccine expressing brachyury based on a modified vaccinia ankara virus vector in chordoma is under way (NCT03595228).

### 4.3. Immune Checkpoint Inhibitor

#### 4.3.1. PD-1/PD-L1

PD-1/PD-L1 ICIs are the most commonly used drugs in chordoma patients. Multiple studies have indicated that high expression of PD-L1 in chordoma tissues and PD-L1-positive tumors are significantly related to advanced chordoma and the prevalence of TILs [46,47,48], and ICIs targeting PD-L1 have shown obvious clinical activities against advanced chordoma [49]. Six of the twelve studies focused on PD-L1 expressing chordoma with seven patients [28,32,34,35,36,37]. Two of the three patients with PD-L1 negative chordoma achieved PR according to RECIST, and one patient experienced a rapid major clinical improvement for nine months [28,34,36]. Two of the four patients with PD-L1 positive chordoma achieved CR or PR according to RECIST, including one patient who experienced a clear tumor bulk reduction and clinical improvement for six months, and the other patient who achieved CR at lung metastases for more than two years [28,32,34,35]. These results suggest that baseline expression of PD-L1 is not a clear predictor of response to immunotherapy for chordoma, and patients with PD-L1-negative chordoma can also benefit from PD-L1/PD-1 ICIs [36,37]. Notably, all patients who were only observed with PR according to RECIST or impressive clinical improvement were assessed for the PD-L1 status [28,32,34,35]. Information on the PD-L1 status in chordoma patients with PD or SD is required to further confirm this conclusion.

Avelumab is the first human anti-PD-L1 IgG1 antibody with a natural Fc region, allowing it to induce antibody-dependent cell-mediated cytotoxicity (ADCC). Additionally, avelumab-mediated ADCC selectively targets and kills cancer stem cells in chordoma [13,50,51]. Its capacity to facilitate ADCC may provide avelumab with a distinct advantage over other PD-1/PD-L1 ICIs when combined with drugs that stimulate NK cell activation or with the adoptive NK cell therapy [22].

#### 4.3.2. CTLA-4

CTLA-4, also known as CD152, is expressed primarily by T cells and mainly controls the early stages of T cell activation. A study reported that CTLA-4 was positively expressed in all TILs and tumor cell cytoplasm, and that high expression of CTLA-4 in tumor tissues was related to a worse prognosis, and CTLA-4 of TIL was only associated with PFS [52]. We found that CTLA-4 ICIs were only used in combination therapy in chordoma. Tremelimumab plus durvalumab and ipilimumab combined with nivolumab are two common regimens for CTLA-4 immune checkpoint combinations. CTLA-4 antibody treatment may be more effective as a combination therapy instead of monotherapy [47]. The reason may be that different immune checkpoints act differently. PD-1 principally modulates the activity of effector T cells in tumors, whereas CTLA-4 mainly regulates the activation of T cells. Given the significance of B cell and NK cell in tumor microenvironment [53,54], the regulation of CTLA-4 takes place more narrowly as compared to PD-1, and it enjoys exclusive expression on T cells [55]. Colitis was the most frequent AEs in combination therapy with CTLA-4 [25,37]. Given that higher toxicity was observed in ICI combination therapy [55], clinicians should properly manage AEs when administering a CTLA-4 immune checkpoint combination.

The best of our acknowledge, there are no large randomized controlled trials that indicate CTLA-4 plus PD-1/PD-L1 combination therapy to be superior to PD-1/PD-L1 monotherapy. Multiple clinical trials are under way to investigate the efficacy of CTLA-4 combined other ICIs (NCT02834013, NCT04416568 and EudraCT2020-002821-28).

#### 4.3.3. Other Targets

To date, clinical ICIs therapy has been limited to ICIs targeting PD-L1/PD-1, and CTLA-4. Notably, the role of many other immune checkpoints in tumors has been discovered, and they may provide new targets for monotherapy or combination treatments. The effects of many immune checkpoints, including LAG3, TIM3, SIRPα, HHLA2, MAGEA4, VISTA and TIGIT [56,57,58,59,60,61,62], have been validated in chordoma, and five of them (TIM3, SIRPα, HHLA2, MAGEA4 and VISTA) were only evaluated in preclinical study. ICIs targeting TIGHT and LAG3 have been included in clinical trials. TIGHT antibody was evaluated in two clinical trials (NCT03886311 and NCT05286801), and one clinical trial on LAG3 antibody is ongoing (NCT03623854).

#### 4.3.4. Prediction of ICIs Response

By distinguishing responders and non-responders, biomarkers that evaluate the effectiveness of ICIs can guide therapy selection [55]. Although PD-L1 expression is regarded as a definitive predictor of ICIs [54], there are exceptions in chordoma [63]. Ample research has demonstrated that PD-L1 expression is not an effective indicator of chordoma immunotherapy [28,36,37,64]. Growing evidence has shown that immune inflamed tumors and microenvironmental tumor-infiltrating lymphocytes (TILs) levels are related with the immunological response to ICIs treatment [35,55,65,66,67].

Emerging evidence indicates that microsatellite instability (MSI) is related to ICIs response in malignancies [68,69,70]. Klinger et al. reported MSI in 50% of the tested chordoma samples, and MSI could be used to predict the response to Pembrolizumab [49,63]. It is primarily important to note that high MSI is a subset of high tumor mutational burden (TMB) [71], meaning that tumors with high MSI do not exclusively have high TMB.

In a case report, the author discussed a patient with metastatic chordoma who achieved PR according to RECIST and had a PFS duration of 9.3 months [36]. Although the TMB was low, he was determined to have the PBRM1 A1209fs mutation. PBRM1 is a gene on chromosome 3 that encodes a component of the PBAF form of the SWI/SNF chromatin remodeling complex. Mutations in the PBRM1 gene are generally associated with ICIs response [36,72].

INI1, also known as SMARCB1, encodes an important core subunit of the SWI/SNF complex, the deficiency of which is associated with poorly differentiated chordoma [35]. More recent research has demonstrated that many INI1-negative tumors are infiltrated by immune cells and express PD-L1, suggesting that these tumors may be more susceptible to ICIs therapy [73,74,75,76]. Two case reports associated with INI-negative tumors included in this review article also support this viewpoint [32,35]. A clinical trial for INI-negative malignancies is underway to explore whether INI1 loss predicts tumor response to ICIs (NCT04416568).

### 4.4. Cell Therapy

The deficiency of HLA class I antigens contributes to tumor escape from immunosurveillance and the mechanism of resistance to T-cell based immunotherapy. However, the proportion of chordomas with defective expression of HLA class I antigen is high. Given the current interest in the use of vaccines to enhance the T cell effect and ICIs, how to treat chordoma with this feature is particularly important [76]. Chimeric antigen receptor (CAR)-T cells were found to exert antitumor effects in a non-HLA restricted manner [76], which may provide an alternative immunotherapy strategy for chordoma patients with HLA class I dysregulation [14]. In the recent year, preclinical studies exploring cell therapy, such as the CAR-T cells and modified NK cells, have emerged in chordoma.

Chondroitin sulfate proteoglycan 4 (CSPG4), also known as HMW-MAA [76], is overexpressed in chordoma and associated with disease metastasis and risk of death. Targeting CSPG4 may be a potential treatment modality for chordoma [76]. Bearded et al. manufactured a second-generation CAR monoclonal antibody against CSPG4 and successfully targeted CSPG4 in several tumor types [44].

Cancer-associated fibroblasts (CAFs) in the microenvironment of chordoma overexpress B7H3, which induces a suppressive immune microenvironment and promotes tumor propagation [57]. A preclinical study showed that B7-H3-targeted CAR-T cells effectively inhibited tumor sphere development in chordoma [76]. Taken together, these studies provide preclinical evidence for phase I clinical trials of anti-CSPG4 and B7-H3 CAR-modified T-cells in chordoma [44].

Two preclinical studies investigated engineered high-affinity NK cells in chordoma. One was modified to carry a high-affinity allele of CD16 and endogenously express IL-2 [50], and the other was modified to carry PD-L1, a specific chimeric antigen receptor. The modified NK cells improve the ADCC action on chordoma cells when combined with ADCC-mediating agents, such as avelumab [13,50]. These studies provide preclinical evidence for follow-up clinical trials of adoptive NK cell combination therapies (Figure 3).

## 5. Limitations

There were some limitations to our study. We included case reports in the systemic review due to the scarcity of accessible data and the rarity of chordoma. Case reports may place too much emphasis on the final result due to the lack of strong evidence. We also excluded non-English publications, and we were unable to get access to some research. All of these defects may increase selection bias. To obtain the optimal immunotherapy regimen in chordoma patients, large prospective randomized clinical trials of chordoma immunotherapy are important and necessary.

## 6. Conclusions

Few clinical data are available to demonstrate the advantages of CTLA-4 ICIs combined with PD-1/PD-L1 ICIs over monotherapy in treating chordoma, and ICIs combination therapy shows more toxicity than monotherapy. Anti-PD-1/PD-L1 ICIs monotherapy is recommended as an immunotherapy in patients with advanced chordoma, which may even bring benefits to PD-L1-negative patients. The Brachyury vaccine has shown good safety in chordoma patients, and future clinical trials should focus on how to improve the efficacy of the Brachyury vaccine. The use of immunomodulatory agents is a promising therapeutic option, though additional clinical trials are required to validate their safety and effectiveness. RECIST may not be an appropriate criterion for assessing intratumoral immunotherapy medications.

## Figures and Tables

**Figure 1 cancers-15-00264-f001:**
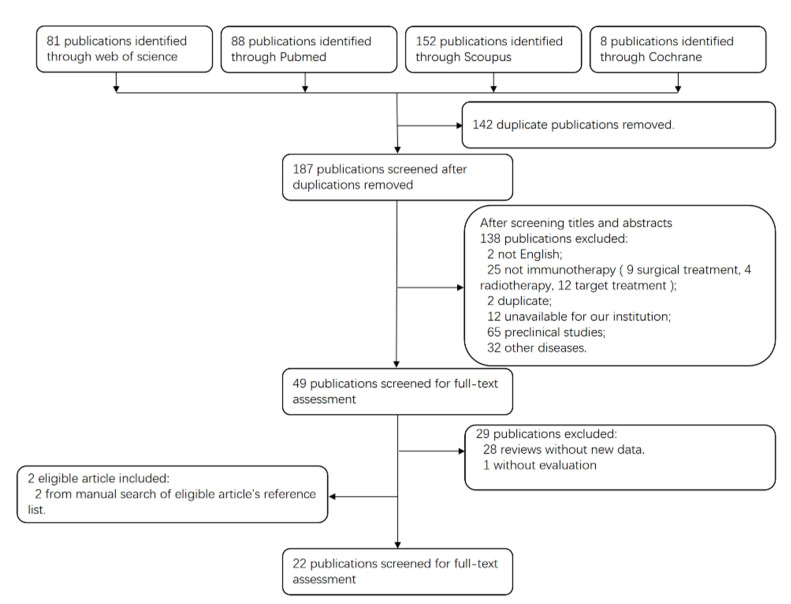
PRISMA flow diagram of the study selection process.

**Figure 2 cancers-15-00264-f002:**
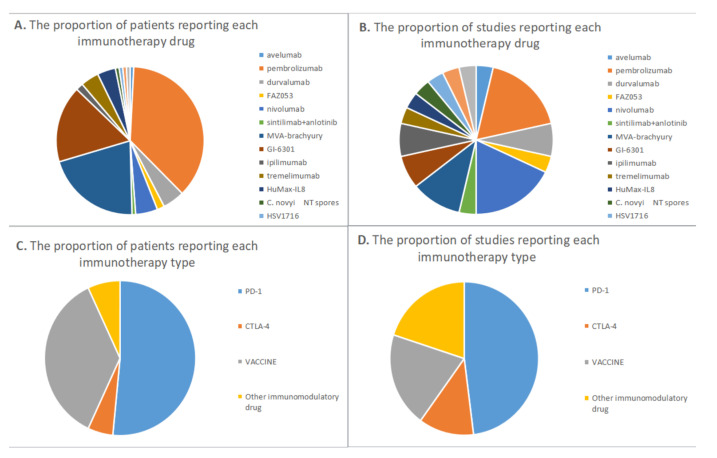
Of the included studies, the proportion of patients (**A**) and studies (**B**) reporting each immunotherapy drug, and the propotion of patients (**C**) and studies (**D**) reporting each immunotherapy type.

**Figure 3 cancers-15-00264-f003:**
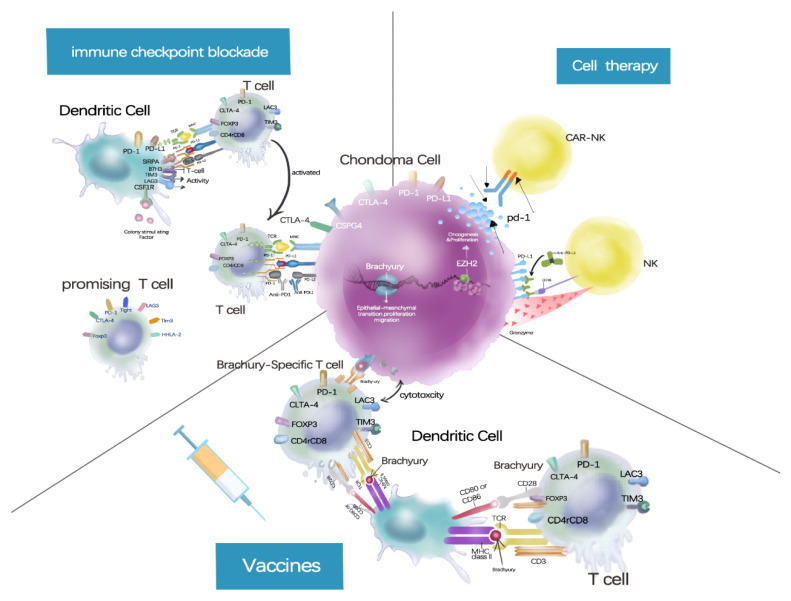
*Vaccines*, vaccines present brachyury antigens to dendritic cells, generating brachyury-specific T cells to kill tumor cells; *Immune checkpoint blockade*, immune checkpoints inhibitors enhance the killing effect of immune cells to tumors by targeting PD-1, PD-L1, CTLA-4 and other targets; *Cell therapy*, Immune cells modified with chimeric antigen receptors can recognize tumor antigens and kill tumors.

**Table 1 cancers-15-00264-t001:** Immunotherapy of chordomas with PD-1/PD-L1 immune checkpoint inhibitor.

Year	Study Design	Levels of Evidence	Sample Size	Tumor Site	Treatment History	Drug	Median Treatment Time (m)	AEs	Choi’s Criteria	RECIST/irRECIST	Median PFS (m)	Median OS (m)
022	case series	IV	9	2 clival 5 spine, and 10 sacrum,	radiotherapy	pembrolizumab 200 mg	12	dermatologic (n = 2), endocrine (n = 2), or Sicca syndrome related (n = 2). grade 3 myocarditis and myositis (n = 1), grade 4 potentially attributable pneumonitis versus infectious reaction (n = 1)		1CR,6SD,2PR,	14 (95% CI, 5–17)	1-year OS was 87%
			2			FAZ053				2SD		
			1			Nivolumab + bempegaldesleukin				PD		
2021	phase2	V	34			pembrolizumab 200mg		The side effect profile of pembrolizumab was similar to other tumor type		3PR	6.6	not reached
2022	case report	V	1	clival	Surgery and radiotherapy	Pembrolizumab (200mg)	3	hypotension, severe fatigue, and dyspnea				
2020	case report	V	1	sacrum	surgery	pembrolizumab		grades 1–2, liver function and hyperglycemia		PR	9.3	
2017	case report	V	1	spine	Surgery and radiotherapy	Pembrolizumab 200 mg		vitiligo			controlled for 6 months	
			1	clival	Surgery and radiotherapy	Nivolumab 3 mg/kg					controlled for 9 months	
2019	case report	V	1	sacrum	radiotherapy	nivolumab, followed by nivolumab + ipilimumab					>4	
2017	case series	IV	2	clival	radiotherapy	pazopanib + nivolumab					161	
2021	case report	V	1	clival		nivolumab				PR		
2021	case report	V	1	sacrum	Surgery and radiotherapy	sintilimab + anlotinib		ICIs-related myocarditis				<1
2017	phase1	V	1			avelumab (1,3,10,20mg/kg)		grade1/grade2				
2019	case series	IV	1			Durvalumab		Sicca/Sjögren’s syndrome				
2022	phase2	V	5			durvalumab + tremelimumab		Colitis, pneumonitis, abdominal pain, myocarditis		1PR+CR,3SD,1PD	13.57 (2.76, 17.81)	

Abbreviations: NR, not reported; PR, partial response; SD, stable disease; PD, progressive disease.

**Table 2 cancers-15-00264-t002:** Immunotherapy of chordomas with CTLA-4 immune checkpoint inhibitor.

Year	Study Design	Levels of Evidence	Sample Size	Tumor Site	Treatment History	Drug	Median Treatment Time (m)	AEs	Choi’s Criteria	RECIST/irRECIST	Median PFS (m)	Median OS (m)
22	case series	IV	5			Durvalumab + tremelimumab	12	Colitis, pneumonitis, abdominal pain, myocarditis		1PD,3SD,1PR		
21	phase1/2	V	1			INT230-6 + ipilimumab		grade1/grade2, anemia, colitis				
19	case report	V	1	sacrum	Radiotherapy	Nivolumab, followed by nivolumab + ipilimumab					>4	

Abbreviations: NR, not reported; PR, partial response; SD, stable disease; PD, progressive disease.

**Table 3 cancers-15-00264-t003:** Immunotherapy of chordomas with Brachuyury vaccine.

Year	Study Design	Levels of Evidence	Sample Size	Tumor Site	Treatment History	Drug	Median Treatment Time (m)	AEs	Choi’s Criteria	RECIST/irRECIST	Median PFS (m)	Median OS (m)
20	phase1	IV	3			MVA-brachyury s.c., 8 × 10^8^ infectious units (IU), followed by FPV-brachyury s.c., 1 × 10^9^ IU,	24	grade1/grade2, injection-site reaction, fever, fatigue		2SD, 1PD	one patient for 52 weeks	
20	phase2	III	11	5 clival, 3 spine and 3 sacrum		GI-6301 (yeast-brachyury vaccine) +RT		injection-site reaction, lymphocyte count decreased, fever		5PD,3SD,1 PR,	20.6	37.5
21	phase1	IV	10			MVA-BN-brachyury-TRICOM vaccine (1 × 10^7^, 1 × 10^8^, 1 × 10^9^)		grade1/grade2, injection-site reaction, fever		4SD,5PD,1PR	253 day	
17	phase1	IV	13			MVA-brachyury-TRICOM (5 × 10^8^, 1 × 10^9^, 2 × 10^9^)		grade1/grade2 (fever, diarrhea) injection-site reaction, lymphocyte count decreased, flu-like symptoms, fever, and diarrhea				
15	phase1	IV	11	3 clival, 2 spine and 6 sacrum.	Surgery and radiotherapy	GI-6301 (yeast-brachyury vaccine) (4, 16, 40, and 80 yeast units (YU))		grade1/grade2, injection-site reaction, fever		3PD,1PR,1mixed response, 5SD	8.3	

Abbreviations: NR, not reported; PR, partial response; SD, stable disease; PD, progressive disease.

**Table 4 cancers-15-00264-t004:** Immunotherapy of chordomas with immunomodulatory drugs.

Year	Study Design	Levels of Evidence	Sample Size	Tumor Site	Treatment History	Drug	Median Treatment Time (m)	AEs	Choi’s Criteria	RECIST/irRECIST	Median PFS (m)	Median OS(m)
2019	phase1	V	5			HuMax-IL8 (4, 8, 16, 32mg/kg)	6	constipation (33.3%), nausea (26.7%) and anemia (26.7%)		SD = 4, PD = 1	5.5	
2018	phase1	V	1			PG545 150mg		Hypertension, Chills, Fatigue		SD		
2021	phase1	V	1			C. novyi NT spores 1 × 10^6^		Pyrexia		SD		
2017	case series	IV	1	clival	Surgery and radiotherapy	MVX-ONCO-1					controlled >19 months	
2017	phase1	V	1	clival	radiotherapy	HSV1716 (2 × 10^6^ i.u.)		fever, chills, anemia and leukopenia		SD	14day	2.5

Abbreviations: NR, not reported; PR, partial response; SD, stable disease; PD, progressive disease.

**Table 5 cancers-15-00264-t005:** Clinical trials programs of chordomas in progress.

Clinical Trial	Trial registration Number	Phase	Medical Condition:	Sites	Status
Talimogene Laherparepvec, Nivolumab and Trabectedin for Sarcoma	NCT03886311	2	Talimogene Laherparepvec, Nivolumab, Trabectedin	USA	Recruiting
Nivolumab and Ipilimumab in Treating Patients With Rare Tumors	NCT02834013	2	Ipilimumab, Nivolumab	USA	Recruiting
Study of Nivolumab and Ipilimumab in Children and Young Adults With INI1-Negative Cancers	NCT04416568	2	Ipilimumab, Nivolumab	USA	Recruiting
Multi-Arm Study to Test the Efficacy of Immunotherapeutic Agents in Multiple Sarcoma Subtypes	NCT02815995	2	Durvalumab, Tremelimumab	USA	Active, not recruiting
A randomised, comparative, prospective, multicentre study of the efficacy of nivolumab + ipilimumab versus pazopanib alone in patients with metastatic or unresectable advanced sarcoma of rare subtype	EudraCT2020-002821-28	2	nivolumab + ipilimumab versus pazopanib	France	Ongoing
A randomized phase II study of Durvalumab (MEDI4736) and Tremelimumab compared to doxorubicin in patients with advanced or metastatic soft tissue sarcoma.	EudraCT 2016-004750-15	2	Durvalumab (MEDI4736) and Tremelimumab compared to doxorubicin	Germany	Ongoing
Multi-Arm Study to Test the Efficacy of Immunotherapeutic Agents in Multiple Sarcoma Subtypes	NCT02815995	2	Durvalumab, Tremelimumab	USA	Active, not recruiting
Nivolumab and Relatlimab in Treating Participants With Advanced Chordoma	NCT03623854	2	Nivolumab, Relatlimab	USA	Recruiting
Tiragolumab and Atezolizumab for the Treatment of Relapsed or Refractory SMARCB1 or SMARCA4 Deficient Tumors	NCT05286801	2	Atezolizumab, Tiragolumab	USA	Recruiting
Phase II trial of the immune checkpoint inhibitor nivolumab in patients with select rare CNS cancers	NCT03173950	2	nivolumab	USA	Recruiting
Phase I safety study of stereotactic radiosurgery with concurrent and adjuvant PD-1 antibody nivolumab in subjects with recurrent or advanced chordoma	NCT02989636	1	nivolumab	USA	Recruiting
TAEK-VAC-HerBy Vaccine for Brachyury and HER2 Expressing Cancer	NCT04246671	1	TAEK-VAC-HerBy Vaccine		Recruiting
BN Brachyury and Radiation in Chordoma	NCT03595228	2	BN-Brachyury plus radiation		
A Study of FAZ053 Single Agent and in Combination with PDR001 in Patients With Advanced Malignancies.	NCT02936102	1	FAZ053 PDR001		Completed Active, not recruiting
Nivolumab (Opdivo^®^) Plus ABI-009 (Nab-rapamycin) for Advanced Sarcoma and Certain Cancers	NCT03190174	2	Nab-Rapamycin Nivolumab		Completed

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
