# Peer review of "Immunotherapy as a Promising Option for the Treatment of Advanced Chordoma: A Systemic Review"

_cancers, 2022, doi:10.3390/cancers15010264_

Round 1

Reviewer 1 Report

Xiang Wang and colleagues have performed a literature review in order to evaluate all clinical data reporting immunotherapy in the treatment of chordoma. The work is huge in terms of analyses of all results including various immunotherapies, as well as various clinical reports (clinical trials and case reports). Hence, such a project is in fact difficult and has been correctly investigated. A number of tables are available to give an (almost) exhaustive overview of all those clinical data. Considering all those considerations, this article appears of interest to identify efficacy of immunotherapies in relapsed chordoma.

In contrast, the discussion seems to repeat the presentation of the results, inducing a heavy and not useful way of comprehension. Particularly, it is very difficult to discriminate between all presented immunotherapies the one(s) which is (are) more efficient and less toxic. Furthermore, as mentioned by the authors, clinical trials have not been separated from case reports in order to define the best immunotherapy in chordoma. Finally, very (very) few data have been mentioned regarding the clinical status of the disease, and, particularly, the site of the tumor, previous surgery and/or radiotherapy.

Overall, this study is of high interest but requires a better presentation of the data that have been obtained, particularly in the sense to try, as far as possible, a well-done comparison between all tested immunotherapies

Author Response

On behalf of all the contributing authors, I would like to express our sincere appreciations of your letter and reviewers’ constructive comments concerning our article entitled “

Immunotherapy: promising options for the treatment of advanced chordoma – a systemic review

” . We sincerely thank the editor and all reviewers for their valuable feedback that we have used to improve the quality of our manuscript. These comments are all valuable and helpful for improving our article. According to the  reviewer comments, we have made extensive modifications to our manuscript  to make our results convincing. In this revised version, changes to our manuscript were all highlighted within the document by using red-colored text. Point-by-point responses to the nice associate editor and two nice reviewers are listed below this letter. The reviewer comments are laid out below in italicized font and specific concerns have been numbered.

1.In contrast, the discussion seems to repeat the presentation of the results, inducing a heavy and not useful way of comprehension.
I've moved the cancer vaccine, as well as the PD-1 discussion section, to the results section. This makes the discussion section better presented

2.very (very) few data have been mentioned regarding the clinical status of the disease, and, particularly, the site of the tumor, previous surgery and/or radiotherapy.
This is invaluable advice, and I've added the patient's tumor location to the table, as well as the treatment history of the patient undergoing surgery or radiation therapy. Because advanced chordoma is a chordoma in which radiotherapy or surgery has failed, most clinical trials have not looked at the treatment history of patients with chordoma and the location of the tumor, except for case reports.

3.Overall, this study is of high interest but requires a better presentation of the data that have been obtained, particularly in the sense to try, as far as possible, a well-done comparison between all tested immunotherapies.
I think this opinion is very valuable. I have added sections comparing adverse events and clinical outcomes between immunotherapy drugs to present the data more visually.

If there are any other modifications we could make, we would like very much to modify them and we really appreciate your help. We tried our best to improve the manuscript and made some changes marked in red in revised paper which will not influence the content and framework of the paper. We appreciate for Editors/Reviewers’ warm work earnestly, and hope the correction will meet with approval. Once again, thank you very much for your comments and suggestions.

Reviewer 2 Report

Thank you for the opportunity to review the systematic review entitled “immunotherapy: promising options for the treatment of advanced chordoma – a systematic review”.  In this systematic review, the authors synthesize results from 22 studies evaluating the safety and efficacy of immune modulating therapies in the treatment of chordoma. This systematic review is a relevant study with broad appeal to clinicians tasked with managing this challenging and rare disease. 

I only have a few very minor comments.

Introduction and discussion section section would benefit from editing by native English speaker/writer.

Tables need some editing so that data columns align.

Otherwise, great work.

Author Response

On behalf of all the contributing authors, I would like to express our sincere appreciations of your letter and reviewers’ constructive comments concerning our article entitled “

Immunotherapy: promising options for the treatment of advanced chordoma – a systemic review

” . We sincerely thank the editor and all reviewers for their valuable feedback that we have used to improve the quality of our manuscript. These comments are all valuable and helpful for improving our article. According to the  reviewer comments, we have made extensive modifications to our manuscript  to make our results convincing. In this revised version, changes to our manuscript were all highlighted within the document by using red-colored text. Point-by-point responses to the nice associate editor and two nice reviewers are listed below this letter. The reviewer comments are laid out below in italicized font and specific concerns have been numbered.

1.Introduction and discussion section section would benefit from editing by native English speaker/writer.

Many thanks to the reviewer for their suggestions, the paper has been submitted to the native English for editing

2.Tables need some editing so that data columns align.

All tables have been aligned and new content has been added

If there are any other modifications we could make, we would like very much to modify them and we really appreciate your help. We tried our best to improve the manuscript and made some changes marked in red in revised paper which will not influence the content and framework of the paper. We appreciate for Editors/Reviewers’ warm work earnestly, and hope the correction will meet with approval. Once again, thank you very much for your comments and suggestions.
